# Post-Transcriptional Regulation of Molecular Determinants during Cardiogenesis

**DOI:** 10.3390/ijms23052839

**Published:** 2022-03-04

**Authors:** Estefania Lozano-Velasco, Carlos Garcia-Padilla, Maria del Mar Muñoz-Gallardo, Francisco Jose Martinez-Amaro, Sheila Caño-Carrillo, Juan Manuel Castillo-Casas, Cristina Sanchez-Fernandez, Amelia E. Aranega, Diego Franco

**Affiliations:** 1Cardiovascular Development Group, Department of Experimental Biology, University of Jaen, 23071 Jaen, Spain; evelasco@ujaen.es (E.L.-V.); cgp00013@red.ujaen.es (C.G.-P.); mmmg0012@red.ujaen.es (M.d.M.M.-G.); fmamaro@ujaen.es (F.J.M.-A.); scano@ujaen.es (S.C.-C.); jmcc0028@red.ujaen.es (J.M.C.-C.); csfernan@ujaen.es (C.S.-F.); aaranega@ujaen.es (A.E.A.); 2Fundación Medina, 18007 Granada, Spain; 3Department of Anatomy, Embryology and Zoology, School of Medicine, University of Extremadura, 06006 Badajoz, Spain

**Keywords:** cardiac development, transcriptional regulation, microRNAs, lncRNAs

## Abstract

Cardiovascular development is initiated soon after gastrulation as bilateral precardiac mesoderm is progressively symmetrically determined at both sides of the developing embryo. The precardiac mesoderm subsequently fused at the embryonic midline constituting an embryonic linear heart tube. As development progress, the embryonic heart displays the first sign of left-right asymmetric morphology by the invariably rightward looping of the initial heart tube and prospective embryonic ventricular and atrial chambers emerged. As cardiac development progresses, the atrial and ventricular chambers enlarged and distinct left and right compartments emerge as consequence of the formation of the interatrial and interventricular septa, respectively. The last steps of cardiac morphogenesis are represented by the completion of atrial and ventricular septation, resulting in the configuration of a double circuitry with distinct systemic and pulmonary chambers, each of them with distinct inlets and outlets connections. Over the last decade, our understanding of the contribution of multiple growth factor signaling cascades such as Tgf-beta, Bmp and Wnt signaling as well as of transcriptional regulators to cardiac morphogenesis have greatly enlarged. Recently, a novel layer of complexity has emerged with the discovery of non-coding RNAs, particularly microRNAs and lncRNAs. Herein, we provide a state-of-the-art review of the contribution of non-coding RNAs during cardiac development. microRNAs and lncRNAs have been reported to functional modulate all stages of cardiac morphogenesis, spanning from lateral plate mesoderm formation to outflow tract septation, by modulating major growth factor signaling pathways as well as those transcriptional regulators involved in cardiac development.

## 1. Growth Factor Signalling in Cardiac Morphogenesis

Specification, differentiation and development of cardiac progenitors are subjected to intense paracrine regulation mediated by different growth factors, particularly bone morphogenetic proteins (Bmps), fibroblastic growth factors (Fgfs), transforming growth factors (Tfgs) and Wnt signaling [1,2,3]. Importantly, these growth factor signals also contribute to additional cardiac morphogenetic processes, such as proepicardial/epicardial development [4,5,6,7,8,9], endocardial cushion formation [10,11,12,13,14,15,16] and outflow tract remodeling [14,17,18,19]. In addition, several other signaling pathways are also required for discrete cardiovascular morphogenetic processes, such as Hippo pathway that is fundamental for cardiomyocyte proliferation and organ size determination [20,21,22], and Notch and neuregulin signaling pathways [23,24,25,26,27,28,29] that are required for ventricular morphogenesis and maturation. Evidence of the functional role of non-coding RNAs impacting on these signaling pathways has been recently reported as detailed in the following subheadings.

## 2. Transcriptional Control of Cardiac Morphogenesis

Cardiac morphogenesis is orchestrated by a large number of transcriptional regulators. Precardiac mesoderm is configured by Mesp1 and Mesp2 expression [30,31,32]. Soon thereafter, the cardiogenic cells are characterized by the expression of cardiac enriched transcription factors such as Gata4, Nkx2.5, Mef2c, Srf that start conferring the initial cardiomyocyte properties and providing positional clues to develop the initial cardiac straight tube [33,34,35,36,37,38,39,40,41,42]. Rightward bending is governed by a differential outgrowth regulated by Prrx1 [43] while Pitx2 imprints left-sided positional cues to the linear heart tube [44,45,46,47,48]. Subsequently, as cardiac chambers emerge, distinct members of the T-box family display regional expression, providing transcriptional cues for the formation of the arterial pole by Tbx1 expression [49,50], the prospective atrioventricular canal and thus part of the slow components of the cardiac conduction system by expression of Tbx2 and Tbx3 [51,52], the left ventricle and atrial chambers by Tbx5 [53,54,55,56] and the external epicardial lining by Tbx18 [57,58]. In addition, systemic and pulmonary ventricles are characterized by Hand1 and Hand2 transcriptional inputs [59,60,61,62], while atrial and ventricular chambers are characterized by Coup-TFII [63], Hey1 and Hey2 expression [64,65], respectively. Thus, overall, an intricate transcriptional regulation governs cardiac morphogenesis, providing evidence of a complex and exquisite regulatory network that involves many transcription factors. 

## 3. The Emergence of a Novel Layer of Gene Regulation: Post-Transcriptional Regulation by Non-Coding RNAs

Whereas transcriptional regulation constitutes a major step governing and defining the molecular mechanisms that direct cardiac morphogenesis and cardiovascular cell differentiation, a novel layer of gene regulation is emerging with the discovery of non-coding RNAs. Non-coding RNAs are broadly classified according to their length in two distinct categories, small non-coding RNA if smaller than 200 nucleotides and long non-coding RNAs if larger than 200 nucleotides [66,67]. Small non-coding RNAs are represented by distinct RNA types such as snoRNAs, siRNAs, piRNAs and the most extensively studied and abundantly expressed microRNAs [68,69]. In all cases, these small non-coding RNAs exerts their function as post-transcriptional regulators, as detailed below. On the other hand, long non-coding RNAs represent a wide and large array of non-coding RNAs that can be classified according to the genomic location (promoter associated, sense, antisense, bidirectional, intronic, intergenic, 3′UTR associated, 5’UTR associated lncRNAs) and/or their functional properties [70,71,72]. Importantly, lncRNAs can be located in distinct subcellular compartments [73,74], such as the nucleus and the cytoplasm, exerting distinct functional roles, i.e., at transcriptional and post-transcriptional, respectively.

## 4. Biogenesis and Function of microRNAs and lncRNAs

MicroRNAs represent the most studied subtype of small non-coding RNAs [69,70]. microRNAs display temporal and spatial differential expression in both embryonic and adult tissues, contributing thus to both embryonic development and tissue homeostasis [75]. Impaired expression and/or function of microRNAs also lead to pathological conditions [76,77]. Importantly, microRNAs are highly conserved during evolution, ranging from *C. elegans* to humans. MicroRNAs are encoded in the nucleus, by transcription of precursors microRNA molecules that are normally transcribed by RNA polymerase II. In certain genomic localization, microRNAs are clustered in such a way that the primary transcript contains multiple microRNA precursors, such as in the case of miR-23/miR-27/miR-24 cluster, a transcript thus named pri-miRNA. Pri-miRNA is then processed by RNAses such as Drosha and Dgcr8 to generate distinct pre-miRNA molecules, i.e., pre-miR-23, pre-miR-27 and pre-miR-24, that are subsequently exported to the cytoplasm by exportin-5/Ran protein complex. Within the cytoplasm, the pre-miRNA is processed into a mature microRNA duplex by Dicer RNAse and loaded into the RISC complex in which one of the double-stranded microRNA molecule is degraded. The mature single-stranded microRNA molecule within the RISC complex is able now to scan other RNA molecules for sequence homology of its seed sequence provoking RNA target cleavage, translation repression and/or RNA deadenylation [78,79] In most cases, the final output consequence is a decrease on the miRNA/protein target abundance.

Long non-coding RNAs also display tissue-specific expression during embryogenesis and tissue homeostasis, being their role in pathology also emerging [73,80,81]. However, their tissue expression levels are, on average, 10-fold lower that mRNAs and microRNAs and they are poorly conserved during evolution. LncRNAs are transcribed in the nucleus, in most cases by RNA polymerase II, and display genomic organization similar to coding RNAs, i.e., introns and exons which are processed by alternative splicing. LcnRNAs are, in most cases, 5′ capped and 3′ polyadenylated and are distinctly distributed in the nucleus or the cytoplasm or both subcellular compartments simultaneously [73,82]. Importantly, although their sequence length can, in some cases, excess 10 Kb, the lncRNAs do not contain open reading frames or if they do, they only provide small oligopeptides or polypeptides that, in most cases, it is unclear if they are indeed translated and/or functional. At the functional level, lncRNAs can exert multiple tasks, such as modulation of transcription factor function, alternative splicing and histone modification (acetylation and/or methylation) at the transcriptional level. On other hand at the post-transcriptional level, they can serve as scaffold for protein post-transcriptional modifications (phosphorylation, protein degradation), multimeric protein complex formation or as competing endogenous RNAs for microRNA sponging [73,80,81]. Furthermore, lncRNAs can also be processed to be loaded into extracellular vesicles, such as exosomes, and thus participate in intercellular communication processes [83,84]. 

## 5. Post-Transcriptional Control of Precardiac Mesoderm Formation by ncRNAs

Bmp signaling plays key determinant roles during cardiogenesis, particularly on the early stages of precardiac mesoderm specification [85,86,87]. Different murine mutant models for Bmp2, Bmp4 and Bmp7 have revealed the importance of these growth factors in cardiogenesis. For example, Bmp2 knock-out mice are embryonic lethal at E7.5 displaying an exocoelomic location of the heart and as well as cardiac differentiation defects [88,89,90]. Bmp4 deficient mice showed even an earlier embryonic lethality than Bmp2 null mutants [91]. Bmp4 induces the expression of Brachyury and Nanog, which in turn induces differentiation of the cardiogenic mesoderm while Bmp2 is necessary for the correct expression of two key factors for early cardiogenesis, Nkx2.5 and Gata4 [92,93]. In this context, it is important to highlight that miR-130 has been recently identified as a key molecular regulator of Bmp2-Fgf8 signaling during early cardiac specification. Bmp2 induces while Fgf8 represses, Nkx2.5 and Gata4 in early precardiac mesoderm. Bmp2 also induces miR-130 expression which directly targets Erk1/2 and thus influences Fgf8 expression [94] (Figure 1A,D).

The role of Wnt signaling in early cardiogenesis is still a matter of debate. Several reports pointed out that Wnt signaling is required for cardiac tissue formation emanating from the early mesoderm [95,96,97] while others reported that cardiac specification can occur independent of Wnt/beta-catenin signaling [98]. Overall, these data support the notion of a tight temporal and tissue specific role for Wnt signaling during early cardiogenesis [99,100]. Importantly, robust evidence on the role of Wnt signaling in cardiogenesis have been found during in vitro embryonic stem cell-derived cardiogenesis [101,102] and the implication of microRNA regulation in this setting has been documented.

Wang et al. [103] reported that miR-218 modulates Wnt signaling in mouse cardiac stem cells, by directly targeting sFRP2 that act as negative regulator of Wnt, thus promoting proliferation while inhibiting differentiation (Figure 1D). More recently, Wang et al. [104] reported the functional role of miR-26b regulating cardiomyocyte differentiation by direct interaction with Gsk3beta and Wnt5a while Liu et al. [105] and Qin et al. [106] analyzed the role of miR-19b in P19 mouse carcinoma cells, demonstrating that miR-19b directly targets Wnt1 and indirectly enhanced Gsk3beta and beta-catenin expression (Figure 1D). Lu et al. [107] investigated the consequences of miR-1 in cardiomyocyte commitment from human cardiovascular progenitors and reported that Fzd7 and Frs2 are direct targets of miR-1 (Figure 1D). Overall, these data illustrate the functional role of microRNAs modulating Wnt signalling in cardiogenesis, yet most of these evidence is reported using in vitro experimental models. 

Several lines of evidence reported the role of Tgf-beta signaling in cardiomyogenic commitment [108,109]. While the functional role of Tgf-beta signaling is thus crucial for multiple steps of heart development, our current understanding of the impact of microRNAs regulating this signaling pathway in cardiogenesis is still incipient. No evidence has been reported on the modulation of Tgf-beta signaling in early cardiomyogenic commitment nor during cardiomyocyte proliferation.

While at present there is no experimental evidence that microRNAs can directly target Mesp1 and/or Mesp2, there is indirect evidence of the functional role regulating the activity of these key cardiogenic transcriptional factors. Coppola et al. [110] demonstrated that miR-99a/let-7c are involved in the control of cardiomyogenesis, in part, by altering epigenetic factors. By over-expressing and inhibiting let7c expression in mouse embryonic stem cells, these authors demonstrate that cardiomyogenesis was promoted by mesodermal specification genes such as T/Bra and nodal as well as cardiac differentiation genes such as Mesp1, Nkx2.5 and Tbx5 were upregulated. Importantly, the functional role of let-7c is restricted to the early phase of mesoderm formation. Let7c induced upregulation of these transcription factors by directly targeting the Polycomb complex group protein Ezh2 and thus modifying H3K27me3 marks from the promoters of crucial cardiac transcription factors. On the other hand, miR-99a represses cardiac differentiation via the nucleosome-remodeling factor Smarca5, attenuating the Nodal/Smad2 signaling (Figure 1A,D).

Additional evidence of the functional role of microRNAs in the early stages of precardiac mesoderm specification was reported by Chen et al. [111] since deletion of Dgcr8 microprocessor, a key protein involved in microRNA biogenesis, in Mesp1 cardiovascular progenitor cells lead to dilated cardiomyopathy, due to defective cardiomyocyte differentiation. These authors identified miR-541 as a highly expressed microRNA in early mouse hearts (E9.5) that partly rescued the Dgcr8 deletion Mesp1 cardiovascular progenitor defects by downregulating angiogenic genes. Shen et al. [112] identified miR-322/miR-503 as the most enriched microRNAs in Mesp1 expressing cells. Ectopic expression of miR-322/miR-503 mimicked the endogenous temporal expression of genes driving cardiomyocyte specification while inhibiting neural lineage development (Figure 1A,D). 

The precardiac mesoderm commitment into the cardiogenic lineage is primarily directed by the combinatorial action of several transcription factors such as Gata4, Mef2c and Nkx2.5. At present limited information is available about the regulation of Gata4 by microRNAs during early cardiogenesis. Yao et al. [113] described that miR-200b targets GATA4, modulating thus myosin heavy chain expression (Figure 1A,D). On the other hand, additional evidence is reported during cardiac hypertrophy as Gata4 is modulated by miR-26 [114] and during hypoxia by miR-200 [115]. While there is scarce indirect evidence of microRNAs targeting Nkx2.5 during early stages of cardiogenesis in mice [116], modulation of miR-1 by Nkx2.5 is essential for early cardiogenesis in *Drosophila* [117] (Figure 1A,D). Additional indirect evidence is also reported in mice, including therein miR-1, miR133 and miR-128b, and demonstrating a key role for miR-128b in cardiogenesis [118]. However, it needs to be considered that some controversies are emanating in this context, as the cardiac miRnome of precardiac mesoderm and early committed cardiomyocytes is not significantly altered in absence of Nkx2.5 [119]. As previously mentioned, Coppola et al. [110] reported that let-7c upregulated genes involved in mesoderm specification as well as cardiac differentiation markers such Mesp1, Nkx2.5 and Tbx5. Evidence of Nkx2.5 regulation by these microRNAs, i.e., miR-1 and miR-133 is also reported in other biological contexts [120].

Finally, similarly scarce reports have demonstrated the regulation of Mef2c by microRNAs in the cardiovascular context. Chinchilla et al. [121] firstly reported that miR-27b directly targets Mef2c, while more recently Chen et al. [122] demonstrated that miR-199 inhibition leads to cardiomyocyte differentiation by up-regulating Mef2c expression in embryonic stem cells (Figure 1A,D). On the other hand, additional evidence on Mef2c regulation by microRNAs has been reported during skeletal muscle development, such as miR-449 [123], miR-194 [124], miR-204 [125], miR-1 [126], miR-206 [126] and miR-214 [127], while miR-488 regulates Mef2c in vascular smooth muscle cells [128].

LncRNAs are versatile RNA molecules that can modulate gene expression either at transcriptional or post-transcriptional level depending on their subcellular localization. To date, our understanding of the functional role of lncRNAs during early stages of heart development is still incipient. A fundamental role for *Braveheart* lncRNA has been reported during early precardiac mesoderm commitment. *Braveheart* is required for the activation of a core cardiovascular gene network by acting upstream of Mesp1. Mechanistically, *Braveheart* interacts with SUZ12, a component of polycomb-repressive complex 2 (PRC2), during cardiomyocyte differentiation [129] (Figure 1A,D). Importantly, *Braveheart* administration is sufficient for mesenchymal stem cells to differentiate into cardiogenic lineage in vitro [130]. Curiously, besides its early developmental role, *Braveheart* is also expressed in the adult heart [131,132] and it is regulated by distinct core cardiac transcription factors [131]. Additional evidence on functional role of lncRNAs during early cardiomyogenic lineage specification has been obtained only using in vitro systems. For example, *uc.245* lncRNA over-expression downregulates the expression of several cardiomyogenic-specific molecular markers such as Nkx2.5, Gata4, Mef2c in P19 cells [132,133]. Similarly, *uc.167* overexpression inhibits proliferation while promoting apoptosis in P19 cells and regulates Mef2c expression [134]. However, the molecular mechanisms underlying such effects remains unexplored. Interestingly *OIP5-AS1* lncRNA interacts with Mef2c mRNA promoting myogenic gene expression in vitro using a skeletal muscle cell line [135] (Figure 1A,D).

Besides *Braveheart*, *Moshe* lncRNA, an antisense transcript located upstream of Gata6 locus, has also been reported to fine-tune early heart development. *Moshe* knock-down during cardiogenesis leads to significant repression of Nkx2.5 in cardiac progenitor stages. RNA immunoprecipitation assays demonstrate that *Moshe* activates Nkx2.5 gene expression via direct binding to its promoter region [136] (Figure 1A,D).

*Fendrr*, a lncRNA located in the vicinity of Foxf1 locus and with enhanced expression at the caudal end of the lateral plate mesoderm is essential for heart and body wall development [137]. *Fendrr* knock out mice are embryonic lethal at E12.5 and displayed hypoplasia of the ventricular chamber and interventricular septum. Mechanistically, Mef2c, Gata6 and Nkx2.5 are upregulated in embryonic hearts whereas Gata6, Foxf1, Pitx2 and Irx3 were also upregulated in the caudal end of the lateral plate mesoderm in this mutant (Figure 2D). Regulation of the expression of these transcription factors is mediated by histone modification at their promoters [138].

## 6. Post-Transcriptional Control of Heart Fields Deployment by ncRNAs

Several transcription factors have been identified as key regulators of second heart field deployment, among which the most relevant are Foxh1, Isl1 and Tbx5 as detailed below. Forkhead box protein family are important components of the signaling pathways that instruct cardiogenesis and embryonic heart development [138]. Particularly, Foxh1 acts as a mesodermal specification inductor during gastrulation and, as the development proceeds its expression is restricted to the developing heart [139,140,141]. In this way, Foxh1 modulates Mef2c and Pitx2 transcription factors during the anterior heart field (AHF) formation which is required for right ventricle (RV) and outflow tract (OFT) development during the heart looping stage [141]. Foxh1 systemic mutants are arrested at the cardiac linear heart tube stage [141]. In zebrafish model, it has been demonstrated that Foxh1 is a direct downstream target of let-7b and miR-103/107 microRNA family [142,143] (Figure 1A,D). In the first steps of development, miR-103/107 modulate Foxh1 expression which together with Smad3 are able to modulate Nodal expression in the lateral plate mesoderm. Impaired expression of this gene can disturb left-right signaling genes generating cardiac edema and kidney failure [143]. More recently, it has been demonstrated that Foxh1 represses miR-430, via non-canonical regulation during early development in zebrafish, balancing Nodal signaling which controls early zygotic gene expression [144] (Figure 1A,D). In other contexts, Foxh1 promotes breast cancer cell proliferation through Wnt/β-catenin signaling pathway activation. In line with that, some authors conclude that let-7b may control cell cycle progression at least in part through the downregulation of Foxh1 [142,145].

Isl1 is a LIM-homeodomain transcription factor important for the development of multiple organs during embryogenesis. In particular, it has a crucial role during the secondary heart field (SHF) formation by directing the contribution of progenitor cells to the deployment of the right ventricle (RV), outflow tract (OFT) and the atria [146,147,148] during cardiovascular development. 

Several microRNAs have been reported to modulate *Isl1* expression in this context. At the first steps of cardiac development, miR-128a modulates cardiac progenitor cells differentiation into different cardiomyocytes subpopulations through the modulation of Isl1 as well as such as Nkx2.5, Mef2c, Irx4 and Shox2 [118] (Figure 1A,D). However, whether such regulatory effects are direct or indirect remains to be established. It has also been demonstrated that BMP-signaling regulates the miR-17-92 cluster that modulate the molecular pathway which promotes SHF myocardial differentiation. Moreover, bioinformatics analyses have shown miR-17 and miR-20a seed sequences in the Isl1 3′UTR and those interactions were validated by luciferase assays (Figure 1A,D). These data reveal the mechanism underlying Bmp-regulated OFT myocardial differentiation and indicate that Bmp-regulated miRNA activity is critical for fine-tuning cardiac progenitor genes [149]. 

Regulation of Isl1 by microRNAs has also been reported in distinct cardiac pathological conditions. In vitro experiments of cardiac hypertrophy stimulation with angiotensin II has evidenced that Isl-1^+^, Sca-1^+^, c-kit^+^ porcine cardiac progenitor cells exhibited high levels of Mef2c, Gata4, miR-29a and miR-21 [150]. Moreover, in another pathological context, such as type 2 diabetes mellitus, miR-128-3p targets Isl1 promoting cardiovascular calcification and insulin resistance modulating Wnt pathway [151].

Tbx5 is a T-box transcription factor family member which has a key role during forelimb and cardiac development. The broad spatiotemporal expression of Tbx5 during development is generally conserved in vertebrates, for example, in human hearts. Tbx5 is expressed in the epicardium, myocardium, and endocardium of embryonic and adult hearts [152]. In zebrafish differential microRNAs and mRNAs expression were analyzed in Tbx5 deficient embryos, identifying miR-19a, miR-30, miR-34, miR-190 and miR-21 as differentially expressed microRNAs [153]. Importantly, some of these microRNAs are already known to be involved in cardiac development [154,155,156,157]. For example, miR-19a is essential for correct heart development [154], and, of particular importance is miR-34a, which has been previously described playing a protecting role against cardiac remodeling in a stress situation [155]. Furthermore, miR-30 modulates *Robo1*, which is implicated in heart tube formation [156] and miR-21 could modulate Ndrg4 causing several cardiac defects [157].

Moreover, Tbx5 and miR-218-1 have a correlated expression in mouse cardiomyocytes. In this line, it has been shown that, in zebrafish, Tbx5 and miR-218-1 dysregulation affect early heart development, concretely, miR-218-1 modulation can rescue cardiac defects generated by Tbx5 over expression, acting as a mediator of Tbx5 during cardiogenesis [158]. In addition, Tbx5 has been reported to be directly targeted by miR-200 [159] during cardiogenesis (Figure 1A,D).

Evidence on the regulatory role of lncRNAs modulating SHF transcription factors has been also reported in vitro by Hou et al. [130] demonstrating that *Braveheart* enhances Isl1 expression as well as other cardiac enriched transcription factors such as Nkx2.5, Gata4, Gata6 during the process of transdifferentiation of mesenchymal stem cells into cells with the cardiogenic phenotype.

Importantly, *Moshe* knock-down also increases expression of SHF lineage genes such as Isl1, as well as several transcription factors relevant for subsequent morphogenetic processes such as Hand2, Tbx2, Shox2 and Tbx18. Therefore, these data suggest that *Moshe* is a heart-enriched lncRNA that controls a sophisticated network of cardiogenesis by repressing genes in SHF via Nkx2.5 during cardiac development [136] (Figure 1A,D).

Modulation of Tbx5, another SHF transcription factor has also been reported to be modulated by lncRNAs. *TBX5-AS1:2* lncRNA regulates Tbx5 expression via modulation of its promoter methylation status. Curiously, *TBX5-AS1:2* lncRNA is dysregulated in Tetralogy of Fallot patients, suggesting a plausible role in the onset of this cardiac congenital defect [160]. Additionally, *Tbx5ua*, a lncRNA close to Tbx5 locus is required for proper ventricular wall development, and thus being necessary for embryonic development, since *Tbx5ua* mouse knock-out embryonic lethal [161] (Figure 1A,D)**.** Besides lncRNAs regulating Tbx5, evidence on the transcriptional regulation of Tbx5 on non-coding RNAs has been recently elucidated by Yang et al. [162]. Over 2600 novel lncRNAs were identified as Tbx5-dependent lncRNAs and these authors further identified *RACER* as a novel lncRNAs that modulate the expression of the calcium-handling gene Ryr2 (Figure 1A,D).

## 7. Post-Transcriptional Control of Sidedness and Cardiac Looping by ncRNAs

The establishment of sidedness and thus the configuration of left-right asymmetry is a complex developmental process in which multiple growth factors are involved, such as nodal, lefty-1 and lefty-2, among others [163,164], that ultimately converge into the activation of Pitx2 [165,166,167,168]. Importantly, Pitx2 is dispensable for correct cardiac looping in mice [169,170] and recently Prrx1 transcription factor has emerged as a key player in this developmental process [43].

Our current understanding of the functional role of microRNAs modulating left-right signaling is progressively emerging, yet most of the evidence has been obtained using in vitro models. Several lines of evidence demonstrated that lefties (lefty-1 and lefty-2) are modulated by miR-302, miR-373 miR-430 and miR-427 in human embryonic stem cells (hESCs) [171,172,173] provoking alterations in early hESCs differentiation. Similarly, the role of miR-430 has been confirmed in zebrafish early development, by using target protectors [174], while miR-217 regulates lefty-1 in mouse embryonic stem cells development, modulating thus mesendoderm formation [175]. In the Japanese flounder, squint is modulated by miR-430 [176] (Figure 1B–D).

Similarly, the role of microRNAs in Wnt signaling during cardiogenesis is limited [177] while increasing evidence has been reported in vitro [103,104,105,106], as previously stated. Importantly, in vivo evidence demonstrated that miR-19b overexpression in zebrafish embryos impaired left-right cardiac signaling leading to impaired cardiac development. Beta-catenin was identified as a direct miR-19b, supporting the notion of impaired Wnt signaling in this experimental model [177] (Figure 1B–D)**.**

Pitx2 is a homeobox transcription factor which is expressed quite early in the embryo [167,178,179,180,181]. Pitx2 expression is restricted to the left lateral plate mesoderm of different species and several studies have demonstrated that Pitx2 is important for transmitting positional information from LPM to organ primordia such as heart, lung and gut [182,183,184]. Moreover, during development and adulthood Pitx2 expression remains in the developing heart, predisposing to atrial arrhythmias [167,185,186,187].

Our current understanding of the impact of microRNAs during embryonic Pitx2 expression is still incipient. Wang et al. [185] demonstrated that Pitx2 positively regulates miR-17-92 and miR-106b-25 repressing the sinoatrial node genetic program (Figure 1B–D). On the other hand, several studies have evidenced that Pitx2 modulates the expression of several microRNAs that contribute to Atrial Fibrillation (AF), such as miR-106-25, miR-17-92, miR-29a, miR-200, miR-203, miR-21, miR-208ab, miR-1, miR-26b and miR-106ab [185,188,189,190,191,192]. The lack of Pitx2 expression in the left atrial appendage impairs the expression of a microRNA battery (miR-21, miR-106a, miR-203 and miR-208ab down-regulated, and miR-1, miR-26b, miR-29a, miR-106b and miR-200 up-regulated) through a Wnt8a and Wnt11 modulation, leading to a calcium, sodium and potassium channel remodeling during AF [185,188,189].

Prrx1 is a paired-like homeobox transcription factor that modulates EMT during embryonic development as well as in pathological conditions such as cancer [193]. During embryonic development, EMT is also regulated by other transcription factors such as Snail, Zeb and Twist. Fazilaty et al. [194] evidenced a feedback mechanism among these EMT inductive factors, regulated by microRNAs. Particularly, they reported that Snail1 represses Prrx1 whereas Prrx1 represses Snail1 via miR-15 regulation. Additionally, Nodal signaling in the left LPM downregulates Prrx1 and Snail1 via the upregulation of some microRNAs, such as let7, miR-34, miR-92, miR-124, miR-125, miR-135 and miR-184 [195]. It has been demonstrated that miR-34a and miR-92a modulate Prrx1 expression in a narrow time window (Figure 1B–D). If these miRNAs are deregulated Prrx1 expression is impaired, and consequently, a mesocardia is observed in embryos due to a loss of L/R asymmetry [195].

Regulation of Pitx2 left-right transcription factor by lncRNAs has been reported in two distinct biological contexts. Within early developmental stages, it has been evidenced that the lncRNA Pitx2 locus asymmetric regulated RNA (*Playrr*) and Pitx2 displayed contralateral expression, in other words, while Pitx2 has a left side pattern expression, *Playrr* is expressed in the right side of the embryo. Furthermore, there is a mutual antagonism between them that is regulated by chromosome structure. As Pitx2 expression needs to be dynamic during development, Pitx2 and *Playrr* interactions are tightly regulated to provide a proper Pitx2 dose for gut development [196] (Figure 1B–D)**.** However, no evidence is reported that such interaction also applies to cardiovascular development. In humans, lncRNA Pitx2 adjacent noncoding RNA (PANCR) and Pitx2 are co-expressed during cardiomyocyte differentiation and within human left atrium. However, functional role remains unclear, particularly in atrial fibrillation, a Pitx2-dependent arrhythmogenic defect [197] (Figure 1B–D).

## 8. Post-Transcriptional Control of Proepicardium/Epicardium Formation by ncRNAs

The proepicardium is a transitory structure that contributes to the developing heart by extruding cells into the external surface of the embryonic myocardium, lining it and subsequently experimenting an epithelial-to-mesenchymal transition that promotes these cells to contribute to distinct cardiovascular cell types, particular to the coronary vasculature and the cardiac fibroskeleton [198,199,200,201].

Tgf-beta signaling plays multiple roles during cardiogenesis, including epicardial development [202,203,204,205,206,207,208,209,210]. The role of microRNAs regulating epicardial development has been reported by Brønnum et al. [211], demonstrating enhanced miR-21 expression. In addition, Pontemezzo et al. [212] have shown selective miR-200c downregulation after Tfg-beta administration, influencing therefore the epithelial-to-mesenchymal transition of epicardial cells.

At transcriptional level, three distinct transcription factors have been reported to play essential roles in proepicardium and epicardium development, namely, Tcf21, Tbx18 and Wt1. Tcf21 is a transcription factor of the “helix-loop-helix” (HLH) family that is expressed in different tissues derived from the mesoderm, including therein the epicardium [213,214]. In the epicardium, one of the main functions of Tcf1 is to regulate the EMT and the subsequent differentiation and specification of cells derived from the epicardium (EPDC) towards different cell types [214]. The expression of Tcf21 in this tissue is regulated by various miRNAs, such as let-7 [215,216] or miR-224, which binds directly to the Tcf21 3′UTR [217] (Figure 2A,D). While the impact of Tcf21 regulation by microRNAs has not been reported to date during embryonic development, it is important to highlight that several examples of Tcf21-microRNA modulation has been reported in cardiac physiopathological conditions. In Caucasian and East Asian populations, an SNP (rs12190287) in the 3′UTR region of Tcf21 prevents miR-224 binding, resulting in impaired Tcf21 expression, a condition related to the development of alterations in the ventricular septum of the heart [218]. A deregulation of Tcf21 produced at this SNP has also been associated with the risk of coronary heart disease [219,220,221]. The lack of miR-224 binding generates inadequate cell differentiation because of alterations in activated proteins 1 (AP-1) and platelet-derived growth factor (PDGF) [222]. Furthermore, Tcf21 deregulation causes greater fibroblast development, since this factor acts as an antagonist in the MYOCD-SRF signaling pathway [223]. Another miRNA that regulates Tcf21 is miR-146a. In an Iranian population study, another SNP (rs2910164) is produced in the sequence of the precursor of miR-146a, altering the structure and expression of mature miRNA [224]. 

Tbx18 is involved in the development of the venous pole, the epicardial development and the development of cardiomyocytes of the interventricular septum [51]. Several miRNAs that regulate the expression of this transcription factor have been described, such as miR-1 and miR-185 in chicken [159] or miR-429 in humans [191]. miR-429 [225] and miR-370-3p [226] regulate the expression of Tbx18 during the development of the sinus node and thus the cardiac pacemaker action potential (Figure 2A,D)**.**

Although it has been widely described that Wt1 is a key transcription factor for epicardial development ([227,228,229,230], non-coding RNAs that regulate the expression of Wt1 during cardiogenesis have not yet been identified.

At present, evidence of the regulation proepicardium/epicardium formation transcription factors by lncRNAs has only been reported for Tbx18, as previously mentioned by *Moshe* [136], while evidence for Tcf21 is lacking and for Wt1 has only been reported in other biological contexts [231,232,233] (Figure 2A,D).

## 9. Post-Transcriptional Control of Conduction System Development by ncRNAs

The cardiac conduction system constitutes the electrical wiring of the heart and it is originated from myocardial precursors sleeves located at the junction of the cardiac chambers [234]. Formation of the cardiac conduction system is highly dynamic during development and several transcription factors such as Tbx2, Tbx3, Irx3 and Irx5 have been reported to play fundamental roles during its configuration as detailed below.

Tbx3 is a cardiac transcription factor involved in the development of the sinoatrial node (SAN) and atrioventricular canal (AVC) [235,236]. Several miRNAs that regulate the expression of Tbx3 have been described, such as miR-17-92 and miR-106b-25, which are also regulated by Pitx2c on the left side. Pitx2c acts by positively regulating miR-17-92 and miR-106b-25 on the left side thus suppressing the development of the sinoatrial node, as both miRNAs negatively regulate Tbx3 expression. However, on the right side, there is no expression of Pitx2c, allowing the activation of the Shox2 signaling pathway and subsequently of Tbx3 [237]. Also important is the role played by miR-1 on Tbx3, controlling the differentiation of sinoatrial precursors (Figure 2B,D). In addition to confirming this by computer analysis [238], it was also experimentally demonstrated that the overexpression of miR-1 in mESC-derived cells produces a decrease in the expression of Tbx3, thus reducing the development of SAN [239]. Tbx3 is also involved in the pluripotency of embryonic stem cells [240]. miR-137 acts as a direct target of Tbx3 repressing its expression, since an overexpression of this miRNA produces a significant decrease in cell proliferation [241] (Figure 2B,D). Tbx3 is able to modulate other factors that also intervene in the regulation of differentiation, such as Oct4, Sox2 or Nanog [242,243]. Such regulation is mediated by phosphatidylinositol-3-OH kinase-Akt and mitogen-activated protein kinase pathways, preferentially stimulating Nanog [244]. Therefore, although miR-137 does not have a direct target on Nanog, its expression can be regulated by the action of Tbx3 [241]. For Tbx2 and Irx3, non-coding RNAs that control the expression of Tbx2 and/or Irx3 during cardiogenesis have not been described at the moment, while scarce evidence is reported for Irx5 [245] in cardiac pathological conditions.

In line with the evidence reported during proepicardium formation, scarce evidence is currently available for the functional role of lncRNA regulating transcription factors with key relevant roles during conduction system formation. *TTN-AS1* is enriched during cardiac development and its expression is regulated by TBX2, suggesting a plausible of *TTN-AS1* during cardiac conduction development [246] (Figure 2B,D).

## 10. Post-Transcriptional Control of Chamber Morphogenesis and Valve Development by ncRNAs

Soon after looping, distinct atrial and ventricular cardiac chambers are configured. Within the atrial chambers pectinated muscles are formed while within the ventricular chamber, the embryonic myocardium is initially configured into a trabecular meshwork that subsequently, as development proceeds is replaced by the progressive formation of a compact myocardial layer.

Several growth factors are implicated in different phases of cardiac chamber morphogenesis, in particular, Notch and Neuregulin play a fundamental role in establishing the trabeculated myocardium while Hippo signaling is essential for organ size.

Notch signaling is widely implicated in cardiogenesis, particularly establishing a cross-talk communication between myocardium and endocardium during chamber formation [247,248,249]. Despite the prominent role of Notch signaling during cardiogenesis, no evidence of microRNAs regulation has been reported directly targeting Notch1 in this context.

Neuregulin signal is essential for the correct ventricular chamber trabeculation [250,251,252,253]. To date, there is no evidence of microRNAs regulating neuregulin in the cardiovascular context, yet Sun et al. [254] identified several microRNAs that are modulated by neuregulin administration. Neuregulin administration in murine embryonic stem cells increased miR-296-3p and miR-200c while decreased miR-465b-5p expression. Importantly, manipulation of these microRNAs, i.e., inhibition of miR-296-3p or miR-200c decreased and inhibition of miR-465-5p, respectively, promoted differentiation of embryonic stem cells into the cardiac lineage (Figure 3A,C). 

Hippo signaling is primarily involved during cardiac development by controlling cardiomyocyte proliferation and thus heart size [255,256] and also plays pivotal roles in cardiac regeneration [20,257] and diseases [258,259]. It has been reported that the miRNA cluster miR302–367 negatively regulates the Hippo pathway in vivo, by directly targeting three key components of this pathway, i.e., Last2, Mst1 and Mob1b thus promoting cardiomyocyte proliferation and dedifferentiation. More recently Xie et al. [259] demonstrated that miR-10b also regulated human embryonic stem cell-derived cardiomyocyte proliferation via directly interacting with Last1, which is a major component of the Hippo pathway (Figure 2C,D).

Several transcription factors have been implicated in different phases of cardiac chamber morphogenesis, being particularly important for atrial vs. ventricular chamber development the transcription factors Hey1, Hey2, Irx4, Coup-tfII and Tbx20, while Hand1 and Hand2 are of importance to systemic vs. pulmonary development of the ventricular chambers and Foxm1, Hop, Kfl13 and Srf for the development of compact vs. trabecular myocardium.

Indirect evidence has been reported on the plausible role of miR-34 regulating distinct components of the Notch signaling pathway, including therein Hey2 [260], as well as for miR-128 regulating Irx4 [118], however no direct interactions between microRNAs and Hey1, Hey2, Irx4 or Coup-tfII in the cardiovascular context have been reported to date (Figure 2C,D). Importantly, several reports in other biological contexts have been widely reported [261,262,263,264,265,266,267].

Another key factor for cardiovascular development is Tbx20, which is involved in processes such as the differentiation of cardiac chambers and the proliferation and differentiation of cardiomyocytes [268]. Despite the importance of this factor during cardiogenesis, only indirect evidence for miR-1, miR-25, miR-141, miR-185 and miR-200a has been described distinctly modulating Tbx20 expression during cardiogenesis [159] (Figure 2C,D). 

On the other hand, several reports are available about the functional role of distinct microRNAs modulating the expression of the Hand1 and Hand2 transcription factors. It has been reported that miR-363 downregulates Hand1, suggesting that miR-363 contributes to the generation of a functional left ventricle [269,270], and also Hand1 is modulated by let7 microRNA family members (mmu-let-7a/7d/7e/7f) at different stages in mouse heart development [271]. On the other hand, miR-1 and miR-133a directly target Hand2, balancing proliferation and differentiation during cardiogenesis [272,273,274] (Figure 2C,D).

Klf13 is a transcriptional factor important during cardiogenesis that physically interact with Gata4 [275]. At present two distinct microRNAs have been shown to directly interact with Klf13, miR-147b [276] and miR-125b [277] (Figure 2C,D).

Serum response factor (SRF) is a member of the superfamily MADS-box (MCM1, agamous, deficient, and SRF) of transcriptional factors and is highly expressed through embryonic, fetal and postnatal development of the heart. Srf regulates multiple target genes by the presence in their promoter of serum response elements, such as genes involved in metabolism, extracellular matrix deposition, gene transcription and protein translation [278]. Several microRNAs regulate Srf expression in distinct cardiovascular contexts, such as miR-483-3p in endothelial progenitor cells [279], miR-22 in human umbilical vein endothelial cells [280], miR-23, miR-93, miR-486 in cardiomyocytes [281] and miR-181 in vascular smooth muscle cells [282]. On the other hand, no evidence is reported on the ncRNA regulation of Foxm1 and Hop, despite their crucial role during cardiogenesis [283,284].

In contrast to our understanding of the impact of lncRNA during proepicardium and conduction system development, evidence of the role of lncRNA during chamber formation and valve development is widely emerging. Evidence on the modulation of Bmp2 has been reported in cardiac valves, mediated by H19 lncRNA. H19 administration silenced Notch1 expression by modulating the recruitment of p53 to its promoter and consequently, Bmp2 and Runx2 are downregulated, as they are downstream targets of Notch1 signalling [285] (Figure 3A,C). In addition, Hand2 transcription is modulated by *Upperhand (Uph)* [286], and this lncRNA is essential for proper cardiac development, as systemic deletion caused hypoplasia of the right ventricle and septal defects [287] (Figure 2C,D).

Furthermore, *Cardinal*, a myocardin adjacent lncRNA, regulates SRF-dependent mitogenic gene transcription as reported by Anderson et al. [288], while *MYOSLID*, an SRF-dependent lncRNA, is a direct transcriptional target of SRF and play essential roles regulating role differentiation while blocking proliferation in vascular smooth muscle cells. Curiously decreased *MYOSLID* expression does not affect SRF expression [289] (Figure 2C,D).

Liu et al. [290] described a regulatory network where *HBL1* lncRNA acts as a functional repressor of miR-1, thus inhibiting Tbx20 expression and cardiomyocyte differentiation. Smad4 is also regulated by lncRNAs. Xiao et al. [291] demonstrated that this transcriptional factor is downregulated by miR-204 which can be regulated by *Malat1* lncRNA acting thus as a miR-204 sponge (Figure 2C,D). Additionally, Klf2 is also regulated by lncRNAs but in other biological contexts, such as modulating monocyte adhesion to endothelial cells by *MANTIS* lncRNA [292] or acute myocardial infarction by *MALAT1* [293] (Figure 2C,D).

## 11. Post-Transcriptional Control of Outflow Tract and Atrioventricular Septation by ncRNAs

Septation of the heart involves the formation of distinct muscular septa, such as the primary and secondary atrial septa in the atrial chambers and the interventricular septum in the ventricular chambers, as well as remodeling of the sinoatrial, atrioventricular and conotruncal endocardial cushions eventually leading to the formation of a four-chambered heart. Within these developmental processes, Bmp, Wnt, Tgf-beta and Notch signaling play essential roles. 

Besides their role in early cardiac specification, Bmp2 and Bmp4 also play an essential role in the differentiation of the outflow tract myocardium derived from the secondary cardiac field [18,19,294,295]. In this context, it has been reported that Bmp2 and Bmp4 induced the expression of two microRNAs encoded by the miR17-92 cluster, miR-17 and miR-20a [296]. Seed sequences of both microRNAs recognized the 3′UTRs from two cardiac progenitor genes, Islet1 and Tbx1 [297,298,299]. The absence of Bmp2/Bmp4 expression results in a down-regulation of miR-17 and miR-20a and thus an increase in Islet1 and Tbx1 expression (Figure 3A,C). Gain and/or loss of function assay and dual-luciferase assays have shown that both microRNAs repress the expression of Islet1 and Tbx1. The down-regulation of miR-17 and miR-20a leads to sustained expression of both transcription factors in cardiac progenitors that prevent myocardial OFT differentiation. As a consequence of failure to silence Islet1 and Tbx1, the Bmp2/4 CKO mutants show a reduced and thickened OFT myocardium accompanied by a down-regulation of sarcomeric proteins such as Mhy6 or Mhy7 [296].

Bai et al. [294] demonstrated a pivotal role of Bmp4/7-miR 17-92 cluster in the development of OFT cushions. Bmp4 mutant mice have shown that it is necessary for OFT septation and cushion development [300] (Figure 3A,C). Interestingly, Bmp4 deficiency increases Bmp7 expression, suggesting a possible compensation mechanism and therefore the involvement in the same morphogenetic process. However, Bmp7 deletion does not result in a visible phenotype, demonstrating that its possible role is residual compared to Bmp4 [300]. Functional assays demonstrated that both induce miR-17-92 expression, which is necessary to correct outflow tract cushion development by inducing epithelial to mesenchymal transition (EMT) [294]. The miR-17-92 cluster triggers Vgfa2 mRNA, which represses EMT of the atrioventricular canal (AVC) [301]. Interestingly, SHF Bmp4/Bmp7 deficiency and the miR-17-92 cluster show similar phenotypes, resulting in a defect of EMT and a reduction in cardiac neural crest entry, with the consequent patent trunk arteriosus showing a function linked in the development of the outflow tract [300].

Functional role of Wnt signaling governing valve formation is extensively documented [302,303,304,305,306,307,308]. However, no evidence has been reported on the role of microRNAs regulating Wnt signaling during cardiac valve development yet. There are compiling evidence in cardiac valve pathological conditions, such as valvular calcification by miR-29b [309] and inflammation by miR-27a [310], as well as in other cardiac pathological conditions such as atrial fibrillation [311] regulating Wnt3 by miR-27b and ischemia/reperfusion regulating Wnt/β-catenin signaling pathway by miR-148b [10].

Tgf-beta signaling also plays a pivotal role in endocardial cushion formation [312,313,314,315,316,317,318,319], where indirect evidence for miR-23 and miR-199a was reported by Lagendijk et al. [320] and Bonet et al. [321] during cardiac valve formation, respectively (Figure 2A,D).

Notch signaling is widely implicated during valve formation [322,323,324,325,326,327]. Despite the prominent role of Notch signaling during cardiogenesis, only miR-34 has been reported as a direct modulator of Notch1 in embryonic endocardial cells [261] while additional evidence on the role of microRNAs regulating Notch signaling in aortic valve calcification has been reported for miR-34 and miR-29a [328,329,330] (Figure 3A,C).

Our current understanding of the transcriptional regulation of the cardiac muscular septa is still incipient, whereas several transcription factors such as Klf2, Sox9, Smad4 and Odd1 have been involved in endocardial cushion septation. Kuppel-like factor 2 (Klf2) is a very important molecule involved in angiogenesis and endothelial vascular formation and proliferation [331,332]. To date most of the evidence provided on the regulation of Klf2 by microRNAs are reported in pathological conditions, miR-92 and miR-32 regulate Klf2 expression in endothelial cells in acute myocardial infarction [331,333]. Similar evidence on the transcriptional control of Klf2 regulating microRNA expression has been reported in distinct biological settings, such as shear stress-inducing expression of miR-30 family members [334], miR-23b, miR-145 and miR-143 [332,335] while reducing the expression of miR-126-5p [332]. More recently, Sindi et al. [335] have shown that Klf2 has the ability to induce the presence of miR-181-5p and miR-324-5p in exosomes, reducing the vascular remodeling in pulmonary hypertension.

Sex-determining region Y box (Sox9) is a very important regulator of chondrocyte differentiation [336,337]. In cardiovascular development, it plays a crucial role regulating endocardial cushion formation and septation. In this context, Li et al. [337] demonstrated the existence of a regulation network between Sox9 and miR-1a as this miRNA downregulates Sox9 expression, correlating those findings with the onset of ventricular septal defects (Figure 3A,C)**.** Additionally, in the context of cardiac fibrosis, Cui et al. [336] established a correlation between miR-145 and Sox9 and verified those finding by dual luciferase assay while Cheng et al. [338] reported a direct regulation of Sox9 by miR-30e in an experimental model of myocardial ischemia-reperfusion injury.

Smad4 plays multiple roles in distinct developmental and adult tissues and thus is widely expressed. As part of the Tgf-beta signaling, its role during endocardial cushion formation has been widely documented [339,340]. Regulation of Smad4 by microRNAs have been documented in distinct cardiovascular settings [339,340], such as regulation by miR-34 and miR-122 in cardiac fibrosis [340], by miR-146a-5p in exosomes of human cardiac-resident mesenchymal progenitor cells (CPC) [341], by miR-26a in hypertension-induced myocardial fibrosis [342], yet the only evidence on the regulatory role of microRNAs in cardiovascular development has been established by Dong et al. [343]. These authors observed Smad4 downregulation and miR-144-3p and miR-544 up-regulation correlating with the presence of several cardiovascular defects (Figure 3A,C)**.**

## 12. Post-Transcriptional Control of Aortic Arch Development by ncRNAs

Finally, a critical step in cardiac development requires the exquisite remodeling of the arterial and venous connections to provide appropriate systemic and venous connections and thus the establishment of a closed and double circulatory system. In this context, several transcription factors such as Tbx1, Foxc1, Foxc2, Prrx1, Prrx2 have been involved.

The Tbx1 gene is a member of the T-box transcription factor family which participates in the arrangement of pharyngeal arch endoderm, differentiation and migration of neural crest cells migration and cardiac outflow tract development [344]. During myocardial differentiation of cardiac progenitors in the secondary heart field, Tbx1 is directly downregulated by miR-17-92 cluster under the control of Bmp signaling [149]. Although Tbx1 is currently described as one of the core genes implicated in congenital heart disease (CHD), the molecular mechanisms underlying its role in the pathogenesis of this abnormality is not fully understood [345]. Recent studies, described by Cao et al. [346] showed that miR-144 inhibits cardiomyocyte proliferation and promotes cardiomyocyte apoptosis by targeting the 3′UTR of Tbx1 via regulating Jak2/Stat1 signaling pathway (Figure 3B,C).

The transcription factor Forkhead box C1 (Foxc1) is essential for cell proliferation, migration, invasion, as well as vascular formation and maturation [347,348]. Foxc1 is also implicated in the regulation of early cardiomyogenesis and the functional properties of ESC-derived cardiomyocytes [349]. Although its specific regulation remains largely unknown, some studies described non-coding RNAs as promising posttranscriptional regulators of this factor. In heart development, miR-322/-502 cluster up-regulated cardiac genes, such as Wnt2, Isl1, Tbx20 or Foxc1, involved in vascular endothelial cell differentiation [112] (Figure 3B,C). A negative correlation between Foxc1 and miR-511-3p expression levels has been discovered by Henn et al. [350] in a model for in vivo induction of neoangiogenesis.

Forkhead box C2 (Foxc2) participates in blood and lymphatic vessel development as well as regulating adipose cell metabolism [351]. This transcription factor is a predicted target of miR-199a-5p, a microRNA which has been implicated in cardiac remodeling regulation and ventricular hypertrophy [352]. In varicose vein tissues, the downregulation of this microRNA may promote vascular smooth muscle cells (VSMC) proliferation by upregulating Foxc2 [353].

The paired-related homeobox genes Prrx1 and Prrx2 are highly expressed in mesenchymal tissues throughout development, with the cardiovascular system being one of those with the highest transcript levels of Prrx genes reported. These transcriptional factors are implicated in matrix modulation and in the inhibition of adipogenesis activating TGF-β signaling pathway [354]. Prrx1, and other transcription factors such as Zeb or Snail, also plays an important role in the epithelial to mesenchymal transition (EMT) that occurs during embryonic heart development. In this process, Prrx1 attenuates Snail1 expression through direct activation of the miR-15 family promoting a decrease in stem cell properties [194] (Figure 3B,C).

At present, none of the main transcription factors currently involved in aortic arch development, such as Tbx1, Prrx1, Prrx2, Foxc1 and Foxc2 have been reported to be regulated by lncRNAs. However, there is evidence that the long-non-coding RNA (lncRNA) *LINC00242* competitively binds to miR-141 and promotes Foxc1 expression accelerating the angiogenesis in gastric cancer [355] while in hepatocellular cancer, Foxc1 binds to the upstream region of *HOTAIR*, a lncRNA highly expressed in cardiac tissues that may epigenetically regulate embryonic heart development by recruiting PRC2 [356].

## 13. Conclusions and Future Perspectives

Cardiovascular development is a complex developmental process in which multiple cell types are involved [357]. Over the last decades, our understanding of the cellular contribution to the developing heart has been enormous, particularly dissecting the deployment of distinct heart fields to the nascent cardiac tube [358,359] as well as the extracardiac contributions emanating from the epicardium and the cardiac neural crest [227,360]. Similarly, our understanding of molecular cascades governing these morphogenetic events has been equally impressive, by dissecting the functional role of multiple growth factors and transcription factors during cardiogenesis [357]. Recently, a novel layer of complexity has emerged with the discovery of non-coding RNAs and their impact on the gene regulatory networks that governed cardiac development. In this review, we have summarized our current understanding of the functional role of both microRNAs and lncRNAs at different stages of heart development. It is important to highlight that although our insights on the regulatory roles of these non-coding RNAs is still incipient, there is convincing evidence of their impact on early aspects of heart formation, ranging from early precardiac mesoderm modulation [94] to cardiac chamber formation and aortic arch remodeling [345]. It is important also to highlight in this context that although we are currently lacking evidence of the functional role of microRNAs and lncRNAs in a large number of molecular determinants that play essential roles during cardiogenesis, there is ample evidence in other biological contexts. For example, regulation of microRNAs by Bmp4 has been reported in other biological settings, such as angiogenesis [361] as well as during cardiac valve calcification, i.e., miR-141, miR-486, miR-204, miR-30 [362,363,364,365]. Similarly, a large body of evidence have been reported on the microRNA modulation of Tgf-beta signaling in cardiac physiopathology, particularly on fibrosis [366,367,368,369,370,371,372,373] involving distinct microRNAs, such as miR-34a/miR-93 [366], miR-21 [367], miR-155 [368], miR-27 [369], miR-22 [370], miR-23 [371] and miR-30c [372].

Similar evidence is also applicable for lncRNAs. A recent study reported increased expression of *SNHG6* lncRNA in ventricular septal defects patients while functionally *SNHG6* lncRNA overexpression modulates miR-101 and Wnt/beta signaling, supporting a plausible role in congenital heart disease [373]. Zhao et al. [374] identified that *NORAD* directly binds to miR-590-3p in human umbilical vein endothelial cells and miR-590-3p directly target proangiogenic agents, such as Vegf, Fgf1 and Fgf2. On the other hand, Sun et al., [375], identified that *TUG1* served as a sponge for miR-590 and Fgf1 is a direct target of miR-590. Importantly, *TUG1* expression is increased in AMI tissues and cardiac fibroblasts treated with Tgf-β1. More recently, evidence of Wnt regulated lncRNA *Walras* was reported, demonstrating a role for this lncRNA in cardiomyocyte cytoarchitecture in the context of atrial fibrillation [376]. Overall, these data support the notion that in next coming years we will greatly enlarge our understanding of the impact of microRNAs and lncRNAs, their interactive networks on each and every aspect of cardiac development. Moreover, it is also expected that a novel class of non-coding RNA, i.e., circRNAs, that at present have been scarcely implicated in cardiac development [377], will also pop up.

## Figures and Tables

**Figure 1 ijms-23-02839-f001:**
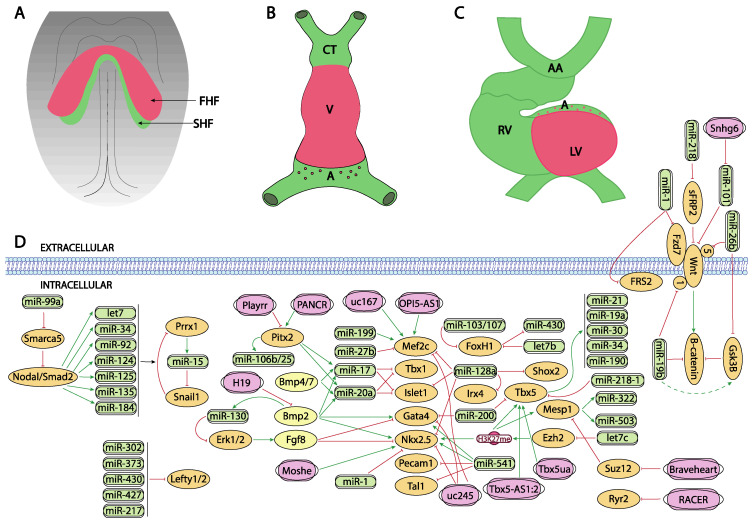
Schematic representation of the main developmental events during the early stages of cardiovascular development, ranging from the configuration of the cardiac crescent (**A**), the early cardiac straight tube (**B**) and the rightward looping (**C**). (**D**) represent the current molecular interaction between microRNAs and lncRNAs with distinct growth factors and transcription factors involved in these developmental processes. FHF, first heart field; SHF, second heart field, CT conotruncus, V, ventricle, A, atrium, RV, right ventricle, LV, left ventricle, AA, aortic arches.

**Figure 2 ijms-23-02839-f002:**
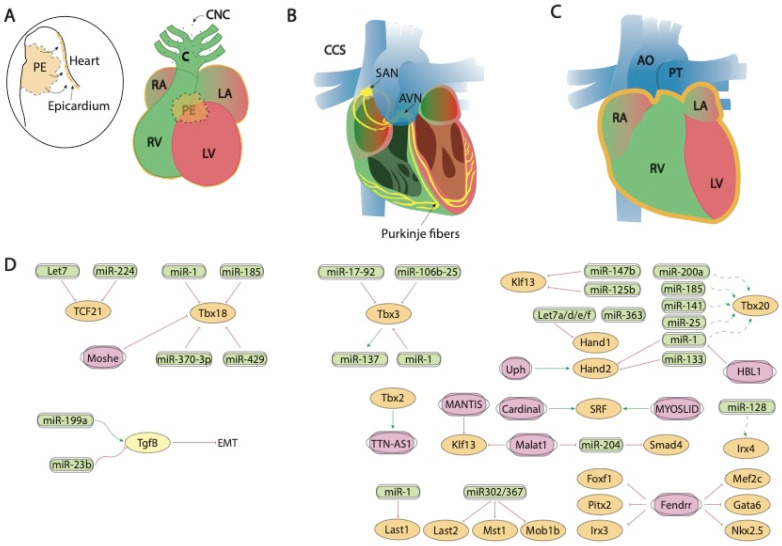
Schematic representation of the mid-developmental events during cardiovascular development, ranging from the epicardial lining of the developing heart and the arrival of the cardiac neural crest cells (**A**), the early configuration of the cardiac conduction system (**B**) and chamber septation process (**C**). (**D**) represent the current molecular interaction between microRNAs and lncRNAs with distinct growth factors and transcription factors involved in these developmental processes. PE, proepicardium, CNC, cardiac neural crest, CCS, cardiac conduction system, RA, right atrium, LA, left atrium, RV, right ventricle, LV, left ventricle, SAN, sinoatrial node, AVN, atrioventricular node, AO, aorta, PT, pulmonary trunk.

**Figure 3 ijms-23-02839-f003:**
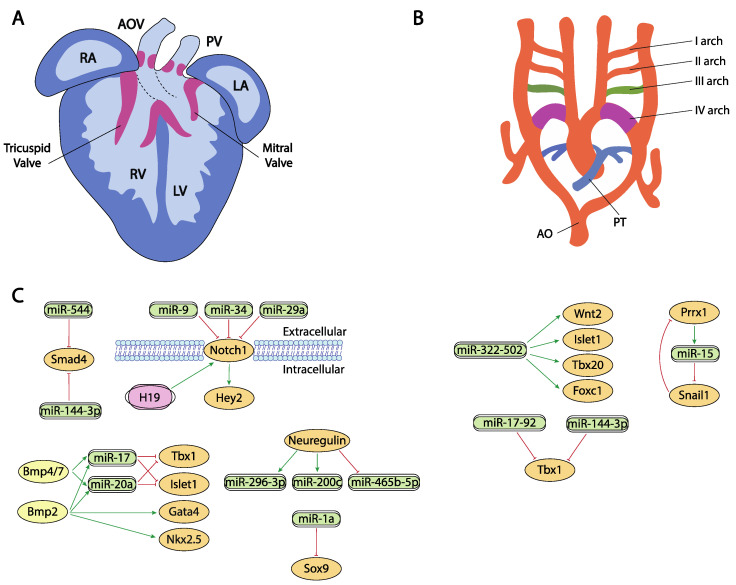
Schematic representation of the cardiac chamber and valvular morphogenesis (**A**) and the configuration and remodeling of the aortic arches (**B**). (**C**) represent the current molecular interaction between microRNAs and lncRNAs with distinct growth factors and transcription factors involved in these developmental processes. RA, right atrium, LA, left atrium, RV, right ventricle, LV, left ventricle, AO, aorta, PT, pulmonary trunk.

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
