# Peer review of "Post-Transcriptional Regulation of Molecular Determinants during Cardiogenesis"

_ijms, 2022, doi:10.3390/ijms23052839_

Round 1
Reviewer 1 Report
In the review “Post-transcriptional regulation of molecular determinants during cardiogenesis” by Lozano-Velasco et al., authors summarized the roles of miRNAs in participating in the regulation of cardiogenesis. The review is well written and comprehensive, covering all the phases of heart development, and the authors did a worthwhile work in reporting all the references published on the topic.
I have only one suggestion to improve the clarity of the review. The figures are surely well done but sometimes it is not clear the link between the main text and the figures. I would suggest using also the letters of the different panels to help the reader (e.g. 1A, 1B,…). In that way, a better visualization of the link between the different events and the molecular interaction could also help (especially for figure 1).
Other minor comments:
- In some lines the figure reference is in bold (e.g. line 236) in other ones no (e.g. line 162): be consistent.
- Line 300: the line has not the format of a title.
Author Response
In the review “Post-transcriptional regulation of molecular determinants during cardiogenesis” by Lozano-Velasco et al., authors summarized the roles of miRNAs in participating in the regulation of cardiogenesis. The review is well written and comprehensive, covering all the phases of heart development, and the authors did a worthwhile work in reporting all the references published on the topic.
First of all, we would like to thank the reviewer for his/her comments which have helped indeed to improve our manuscript.
I have only one suggestion to improve the clarity of the review. The figures are surely well done but sometimes it is not clear the link between the main text and the figures. I would suggest using also the letters of the different panels to help the reader (e.g. 1A, 1B,…). In that way, a better visualization of the link between the different events and the molecular interaction could also help (especially for figure 1).
Following the suggestion of the reviewer, we have improved the figures calls in the main text to ease the reader to follow the review with the help of the different illustrations.
Other minor comments:
- In some lines the figure reference is in bold (e.g. line 236) in other ones no (e.g. line 162): be consistent.
We have modified all the figure references in the text in the revised version of the manuscript
- Line 300: the line has not the format of a title.
We have given the 7th subheading the adequate format as pointed out by the reviewer
Reviewer 2 Report
The review by Lozano-Velasco covers the topic of heart development and regulation by post-transcriptional control, with an emphasis in non-coding RNAs. The review is well-written, very detailed and lengthy. The figures are excellent and make a good job at illustrating the structure of the heart at specific developmental milestones and how non-coding RNAs regulate the expression of the proteins involved. Overall, I think this is a very interesting and detailed compendium of what we know about heart development and post-transcriptional regulation. My only concern (minor) are some grammatical mistakes and sentences oddly formulated peppered throughout the text. This can be easily corrected by careful revision of the English language employed.
Author Response
The review by Lozano-Velasco covers the topic of heart development and regulation by post-transcriptional control, with an emphasis in non-coding RNAs. The review is well-written, very detailed and lengthy. The figures are excellent and make a good job at illustrating the structure of the heart at specific developmental milestones and how non-coding RNAs regulate the expression of the proteins involved. Overall, I think this is a very interesting and detailed compendium of what we know about heart development and post-transcriptional regulation. My only concern (minor) are some grammatical mistakes and sentences oddly formulated peppered throughout the text. This can be easily corrected by careful revision of the English language employed.
First of all, we would like to thank the reviewer for his/her comments which have helped indeed to improve our manuscript. Following the recommendation of the reviewer, the have thoroughly revised the entire manuscript to avoid grammatical mistakes.
Reviewer 3 Report
In this review, Lozano-Velasco and colleagues discuss on the cardiovascular developmental process and on the cellular contribution to cardiogenesis. The authors focus on the molecular cascades governing these events, mainly on the functional involvement of non-coding RNAS (microRNAs and lncRNAs) at different stage of heart development.
The review is valuable to the field. It is well written, concepts are clearly presented and discussed, the diagrams in figures are appropriate and easy to read and understand. I do not have comments on the manuscript other than that.
Author Response
In this review, Lozano-Velasco and colleagues discuss on the cardiovascular developmental process and on the cellular contribution to cardiogenesis. The authors focus on the molecular cascades governing these events, mainly on the functional involvement of non-coding RNAS (microRNAs and lncRNAs) at different stage of heart development.
The review is valuable to the field. It is well written, concepts are clearly presented and discussed, the diagrams in figures are appropriate and easy to read and understand. I do not have comments on the manuscript other than that.
We would like to thank the reviewer for his/her positive comments on our manuscript. We are indeed very happy that he/she has appreciated our work.
Reviewer 4 Report
The manuscript by Lozano-Velasco and colleagues is a comprehensive review describing the state of the art knowledge on the role of non-coding RNAs in cardiac development, with a focus on microRNAs and lncRNAs. The Authors thoroughly describe the complex interactions between these RNAs and the “classic” molecular network acting during cardiac development, with an eye on of the building plan of the heart.
General comment:
The effort of the Authors to present within a single manuscript the complex links between non coding RNAs and the molecular determinants of cardiac development is appreciable. The counter back of this choice is the huge amount of data presented and the excessive length of the review: this can overwhelm the reader (I have been overwhelmed..), and can discourage a reading from the beginning to the end.
A concise comment to the manuscript is that it is well conceptualized by the figures, but not by the format.
Thus, to be acceptable for publication, some editing and a revision of the format are required, as detailed below:
Abstract:
it is definitively too long and unbalanced. Lines 1-31 are devoted to refresh cardiac developmental knowledge to the reader and the topic of the review is only introduced at line 31. Please shorten and reformat it.
Manuscript body:
-The Authors have decided to split microRNAs and lncRNAs contribution in two different parts of the manuscript. Thus, at the mid of the review, the reader has to re-start the developmental round again, encountering the same molecular determinant but different types of non-coding RNAs. I acknowledge that for the writer point of view it is much easier to split these description in different paragraphs. However, from the reader’s point of view, a restart of the story in the mid of the manuscript is quite unsettling. Also, it does not help to go back again to Figures 1-3.
So, the Authors should make an effort in trying to merge the contribution of microRNAs and lncRNAs at each developmental point presented. One possible solution could to split each paragraphs in two subsections, however the Authors could find other means to optimize the text.
- Sometimes no data are available on the developmental function of non-coding RNAs on cardiac TF. The Authors are not discouraged and overcome these limitations by presenting their function in disease (see for example: lines262-271, 292-299, 374-388, 631-638, 708 to 711). I have had the feeling that the Authors are anxious to make us sure that non-coding RNAs are really crucial for the complex biology of TF action, however this is widely recognized nowadays. So, given the huge amount of data already presented for cardiac development, these additional information are excessive and could be highly summarized.
Minor:
- Please pay more attention to the grammar and the style, as there are several endless sentences and several typos mistakes. I have listed below what has come to my attention:
L 51: verb is missing
L 52-53: at present ….recently reported
L 63: emerged
L 72: is involved
L 95: thus , remove.
L 112: they
L 115: displayed
L 120-122: not clear, please edit
L 122-127: 5 lines sentence
L 177: indirectly
L 206: verb is missing
L 208-212: sentence is too long
L 244: . In line with that..
L 327: developing, remove
L 357-361: 6 lines sentence
L 407: is constitute
L 445_446: not clear
L 458-463: 6 lines sentence
L 482-485: this is non clear
L 493: a, should be and
L 538 to 543: too long
L 582: such as
L 642: in, remove
L 701: dynamically during?
L 728-729: modulation..modulated
Author Response
The manuscript by Lozano-Velasco and colleagues is a comprehensive review describing the state of the art knowledge on the role of non-coding RNAs in cardiac development, with a focus on microRNAs and lncRNAs. The Authors thoroughly describe the complex interactions between these RNAs and the “classic” molecular network acting during cardiac development, with an eye on of the building plan of the heart.
General comment:
The effort of the Authors to present within a single manuscript the complex links between non coding RNAs and the molecular determinants of cardiac development is appreciable. The counter back of this choice is the huge amount of data presented and the excessive length of the review: this can overwhelm the reader (I have been overwhelmed..), and can discourage a reading from the beginning to the end.
A concise comment to the manuscript is that it is well conceptualized by the figures, but not by the format.
Thus, to be acceptable for publication, some editing and a revision of the format are required, as detailed below:
First of all, we would like to thank the reviewer for his/her comments which have helped indeed to improve our manuscript.
Abstract:
it is definitively too long and unbalanced. Lines 1-31 are devoted to refresh cardiac developmental knowledge to the reader and the topic of the review is only introduced at line 31. Please shorten and reformat it.
Following the recommendation of the reviewer we have shortened and edited the abstract in the revised version of the manuscript providing a more balanced description.
Manuscript body:
-The Authors have decided to split microRNAs and lncRNAs contribution in two different parts of the manuscript. Thus, at the mid of the review, the reader has to re-start the developmental round again, encountering the same molecular determinant but different types of non-coding RNAs. I acknowledge that for the writer point of view it is much easier to split these description in different paragraphs. However, from the reader’s point of view, a restart of the story in the mid of the manuscript is quite unsettling. Also, it does not help to go back again to Figures 1-3.
So, the Authors should make an effort in trying to merge the contribution of microRNAs and lncRNAs at each developmental point presented. One possible solution could to split each paragraphs in two subsections, however the Authors could find other means to optimize the text.
We would like to thank the reviewer for his/her comment about the format of our review. Following his/her recommendations, we have merged microRNAs and lncRNAs information on each developmental stage.
- Sometimes no data are available on the developmental function of non-coding RNAs on cardiac TF. The Authors are not discouraged and overcome these limitations by presenting their function in disease (see for example: lines262-271, 292-299, 374-388, 631-638, 708 to 711). I have had the feeling that the Authors are anxious to make us sure that non-coding RNAs are really crucial for the complex biology of TF action, however this is widely recognized nowadays. So, given the huge amount of data already presented for cardiac development, these additional information are excessive and could be highly summarized.
Following the recommendation of the reviewer, we have shortened and/or removed this information along the entire manuscript, when applicable.
Minor:
- Please pay more attention to the grammar and the style, as there are several endless sentences and several typos mistakes. I have listed below what has come to my attention:
Following the recommendation of the reviewer we have performed a careful and extensive language revision to correct all the grammatical errors and to avoid any other additional mistakes to make the manuscript more readable.
L 51: verb is missing
Following the recommendation of the reviewer, it has been modified accordingly
L 52-53: at present ….recently reported
Following the recommendation of the reviewer, it has been modified accordingly
L 63: emerged
Following the recommendation of the reviewer, it has been modified accordingly
L 72: is involved
Following the recommendation of the reviewer, it has been modified accordingly
L 95: thus , remove.
Following the recommendation of the reviewer, it has been modified accordingly
L 112: they
Following the recommendation of the reviewer, it has been modified accordingly
L 115: displayed
Following the recommendation of the reviewer, it has been modified accordingly
L 120-122: not clear, please edit
Following the recommendation of the reviewer, we have edited this sentence to make it clearer.
L 122-127: 5 lines sentence
Following the recommendation of the reviewer, we have added some punctuation to make it more readable.
L 177: indirectly
Following the recommendation of the reviewer, it has been modified accordingly
L 206: verb is missing
Following the recommendation of the reviewer, it has been modified accordingly
L 208-212: sentence is too long
Following the recommendation of the reviewer, we have added some punctuation to make it more readable.
L 244: . In line with that..
Following the recommendation of the reviewer, it has been modified accordingly
L 327: developing, remove
Following the recommendation of the reviewer, it has been modified accordingly
L 357-361: 6 lines sentence
Following the recommendation of the reviewer, we have added some punctuation to make it more readable.
L 407: is constitute
Following the recommendation of the reviewer, it has been modified accordingly
L 445_446: not clear
Following the recommendation of the reviewer, we have edited this sentence to make it clearer
L 458-463: 6 lines sentence
Following the recommendation of the reviewer, we have added some punctuation to make it more readable.
L 482-485: this is non clear
Following the recommendation of the reviewer, we have edited this sentence to make it clearer
L 493: a, should be and
Following the recommendation of the reviewer, it has been modified accordingly
L 538 to 543: too long
Following the recommendation of the reviewer, we have added some punctuation to make it more readable.
L 582: such as
Following the recommendation of the reviewer, it has been modified accordingly
L 642: in, remove
Following the recommendation of the reviewer, it has been modified accordingly
L 701: dynamically during?
Following the recommendation of the reviewer, it has been modified accordingly
L 728-729: modulation..modulated
Following the recommendation of the reviewer, it has been modified accordingly
Round 2
Reviewer 4 Report
The manuscript has been widely revised. It is now acceptable for publication in the present form